**communications**

**engineering**

# Energy losses in photovoltaic generators due to wind patterns

Carlos Rossa [1✉]

Previously, in small scale demonstrations, researchers have increased photovoltaic efficiency through cooling by enhancing heat transfer from panels to the air through wind speed. Here I show in the real-world operation of a larger scale photovoltaic generator that increases in wind speed can lead to small but notable energy losses, reflected in the mismatch losses directly derived from the operating voltage of each module. Temperature distribution was measured simultaneously with the operating voltages, alongside the local wind speed and direction. Temperature differences arose from the variable heat transfer throughout the panel, depending on the wind incidence. This affected the operating temperature of each module, consequently affecting their operating voltage and the overall mismatch losses with losses increasing by up to 0.28%. My results suggest that wind patterns cannot be neglected, considering long-term energy estimations and the lifespan of a photovoltaic power plant.

[1] Department of Physics and Mathematics, Climate Physics Group. University of Alcala, Alcala de Henares, Madrid 28801, Spain. ✉email: carlosrossa66y@gmail.com

This paper analyses the energy losses in photovoltaic (PV) generators due to the wind patterns, assessed through the experimental mismatch losses (MML) analysis between PV modules. For monofacial PV generators, MML is understood as the sum of two components: extrinsic, related to the thermal variations induced by the wind, and intrinsic, related to the slight differences in the internal constitution of each module[1]. For example, according to the equation originally proposed for cell-to-cell variations[2], adapted by module-to-module variations[3], the PV generator analysed here has an intrinsic MML of 0.09%[1]. This means that, according to its series/parallel configuration, electrical characteristics and operating conditions dispersion, the minimum expected losses for this generator are 0.09%. In other words, this would be the MML value in the total absence of thermal variations. The intrinsic MML is assumed as constant in the short term, while the extrinsic component largely varies with the wind speed and direction, which can increase the MML roughly three times with respect to the intrinsic value. The thermal behaviour in larger systems follows the fluid mechanics theory for flat plates[4], where the air flux development throughout the PV generator impacts how the temperatures are distributed in it due to the variations of the heat transfer from the surface to the air. This is also affected by the rack/mounting structure, especially for tracker-mounted bifacial modules, as they affect how the incident wind flows throughout the panel and, consequently, how the heat is transferred from the surface to the air. These differences were reported to be up to 14 °C in a previous study[5], with an analysis of a PV generator with an azimuthal tracker, tilted 45°, with similar vertical and horizontal dimensions, whose temperature distribution presents a difference between the two dimensions. The tilt causes an acceleration of the wind incident, increasing the turbulence and reducing the maximum temperature difference $\Delta T$ between two measurement points, which explains the lower $\Delta T$ in the vertical dimension than in the horizontal, as reported by this study. The impact generated by the wind remained unknown until now due to the difficulty in controlling the wind and other climatic variables. This is the reason why the current state-of-art in large PV systems thermal analysis is limited to computational simulation[6,7], as well as wind tunnels with reduced scale panels to simulate a real big PV plant[8,9]. According to these studies, enhancing the convection coefficient with the tilt of the PV panel increases the heat transfer to the air and reduces its temperature. It is well known that cell overheating decreases the overall conversion efficiency of electricity[10,11] and increases the ageing of the PV system[12]. Besides, the assumption that this enhancement and consequent temperature decrease must bring benefits to the overall energy efficiency increase is based on results obtained in experimental analyses of PV/thermal PV (PV/PVT) stand-alone installations[13–22]. In these cases, the interface module/air flux is small to appreciate a full airflow development in natural conditions. In big PV generators exposed to wind patterns, the module temperature depends on its position inside them, as the air flux affects how the module exchanges heat with it. This is the reason why the wind speed increase did not necessarily bring the best PV performance[23,24].

Some studies analysed the impact of the wind in real big PV generators focusing on the energy output[24,25]. However, this masks the effects of air flux variations and consequent temperature differences $\Delta T$—i.e. the difference between the maximum and minimum values of measured temperatures—inside them, which affects the electrical parameters inside each PV module[26,27]. The method of MML—see section Methods—analysis between PV modules used here is accurate enough to see these slight variations due to the wind speed and direction[1]. At first glance, they can represent a value small enough to be neglected. However, neglecting it must represent a significant uncertainty in the bankability of a PV plant, concerning both the project phase, energy production estimations and lifespan due to the thermal stresses induced by the wind patterns.

Here I demonstrated that the wind speed increases the temperature differences, increasing the mismatch and consequent energy losses. Along the same lines, the absence of wind leads to low MML. When evaluated in the long-term, these losses behave similarly, year-by-year, suggesting a connection between them and the local wind patterns. This implies that windy locations may lead to unexpected energy losses. Additional results are included in the Supplementary Material, as Suppl. Figs. S1–S4.

## Results and discussion

**Implications of the transients inside PV modules: temperature and operating voltage differences.** Low wind speeds are sufficient to induce thermal gradients inside PV generators, modules or even inside single cells. These thermal processes are quite dynamic and variable: the simple change in wind direction suffices to change the airflow patterns and, consequently, the temperature differences $\Delta T$. These changes can take place in a few seconds (Suppl. Fig. S1 a, b). $\Delta T$ is positively related to the effective irradiance $G$[28], independently of the period of the year (Suppl. Fig. S1c, d) and the type of mounting of the PV generator[5]. The maximum $\Delta T$ values are likely to occur in windy periods. The operating voltage difference $\Delta V_{OP}$— i.e. the difference between the maximum and minimum values of measured operating voltages—gives a first approach to understanding the overall impact of $\Delta T$ inside the whole PV generator, as both parameters were measured at the same time: $\Delta V_{OP}$ decreases with $G$ increase, maintaining a constant variation for $G \geq 700\ \mathrm{W\ m^{-2}}$, from 0.5 V to 2.0 V, approximately (Suppl. Fig. S1e). This $\Delta V_{OP}$ variation is caused by the thermal variations driven by the wind, which is the reason behind the energy losses. Minimum $\Delta V_{OP}$ values must occur with very low or null wind speeds, while the maximum values relate to higher wind speeds. These variations are in the core of the MML, as will be seen in the next sections, as well as in the section Methods.

The energy losses due to thermal variations were suggested in a previous study to be low[29]. However, they cannot be neglected at all. The high $\Delta V_{OP}$ in low irradiances, as can be seen in Suppl. Fig. S1e occurs due to the shunt resistance dispersions, typically large in Back Surface Field (BSF) multi-crystalline silicon cells[1,30].

**The thermal behaviour from the theoretical point of view.** To understand the thermal behaviour in a PV panel, it can be assumed that the wind flows parallel to it. According to the theory[4], for flat plates, the thermal gradient between the surface and the air strongly depends on the conditions of the thermal boundary layer. Assuming an initial condition in which all the PV panel is at the same temperature, the convection coefficient may be properly described by:

$$h = \frac{k_f \frac{\partial T}{\partial y}\big|_{y=0}}{T_S - T_\infty} \tag{1}$$

where $k_f$ is the heat conduction coefficient from Fourier's law, $\frac{\partial T}{\partial y}\big|_{y=0}$ is the temperature gradient evaluated on the surface of the modules and $T_S$ and $T_\infty$ are the surface and fluid temperatures from Newton's law of cooling, respectively. While the wind flows throughout the panel, both the velocity and thermal boundary layers grow up similarly (Fig. 1a, b) due to viscosity effects inducted by shear stresses. As $T_S - T_\infty$ is constant in any part of the PV panel, the thermal gradients between the surface and the air in the boundary layer decrease while the wind flows throughout the panel. While the entire boundary layer thickens in the x direction, reducing the flow velocity near the surface, a

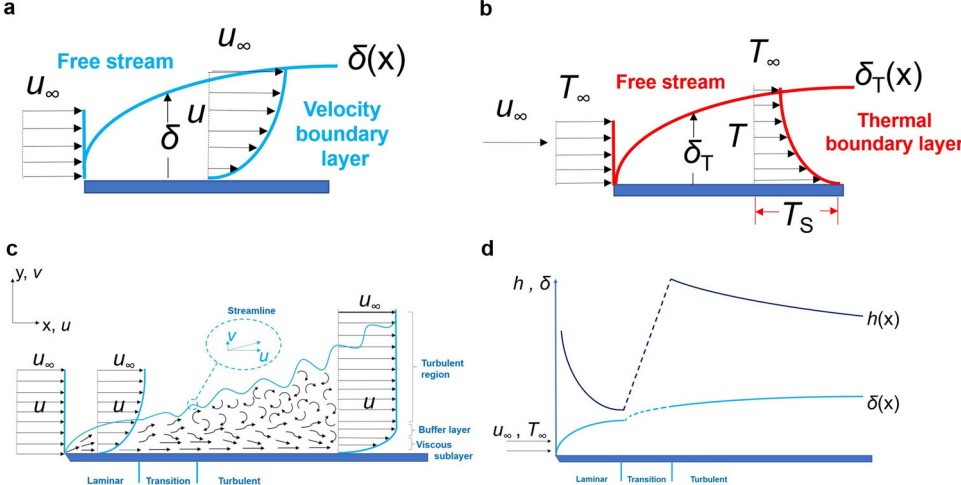

**Fig. 1 Air flux development. a** velocity boundary layer and **b** thermal boundary layer development, parallel to the surface. **c** Air flux development in an isothermal flat plate, used to explain the temperature distribution in a PV generator. **d** variation of the heat transfer coefficient and the velocity boundary layer over the PV panel. **a–d** are adapted from Incropera & DeWitt[4]. $u$ and $v$ represent, respectively, the wind velocity in the $x$ (horizontal) and $y$ (vertical) reference axes, subject to variations inside the thermal boundary layer. $u_\infty$ represents the wind velocity at the free stream. $T$, $T_\infty$ and $T_S$ represent the air/wind temperature inside the thermal boundary layer, the temperature in the free stream and the temperature at the contact surface, respectively. $\delta_T$, $\delta$ and $h$ represent the thermal and velocity boundary layers and the heat coefficient, respectively, where $(x)$ stands for their growth in the $x$ direction.

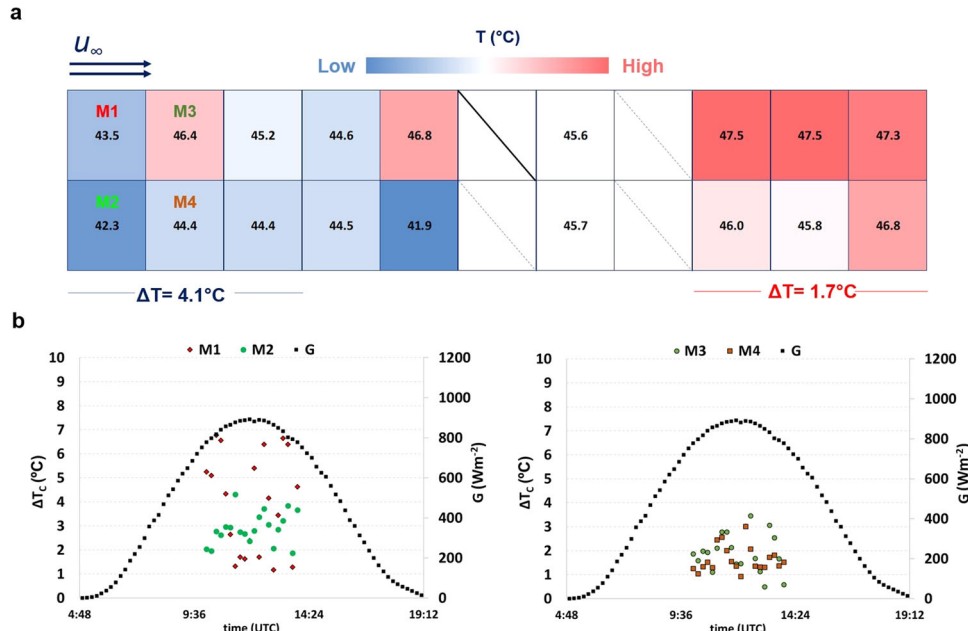

**Fig. 2 Experimental observations of temperature distribution. a** Representation of the back view of the PV generator with the individual temperatures of 18 PV modules, recorded in a single moment, with wind flowing from left to right, parallel to the PV generator. **b** Temperature difference $\Delta T$ inside four modules (M1–M4) during the same sunny day. u and v represent the wind velocity in the $x$ and $y$ reference axes, respectively. $u_\infty$ represents the wind velocity at the free stream.

dampening layer appears under the turbulent zone, which traps the air below it in a viscous sublayer (Fig. 1c). Here, the heat transfer is dominated by conduction, which acts slower than convection, as the thermal energy is transmitted molecule by molecule. The overall heat transfer then decreases (Fig. 1d), and the temperature of the modules remains high in this zone.

Figure 2a shows the experimental observations for a case where the wind flows parallel to the PV generator. These temperature patterns can be observed independently if the generator is delivering or not energy into the grid (Suppl. Fig. S2a): despite the higher temperatures usually observed in $V_{OC}$, as all the sunlight is

converted to heat, the temperature distribution follows the same expected patterns due to the airflow behaviour—the warmest temperatures in the turbulent zone. This reinforces that the thermal behaviour inside PV generators must be dominated by fluid mechanics properties. Other effects like the Joule effect due to the generated current[24] must have a low or depreciable impact to increase the temperatures.

$\Delta T$ tends to be low where the heat transfer is minimal. At the beginning of the flow in an ideal case (laminar), the local heat transfer drops fast (Fig. 1d), possibly due to the colder downwind that supplants the warmest upwind, creating a dampening zone

similar to the occurred in the turbulent flux. The colder air is more viscous, which decreases the heat transfer from the panel to the air, trapping the heat inside this layer. Thus, local $\Delta T$ decreases just at the beginning of the flow: it can be seen, for example, how $\Delta T$ evaluated inside the four modules at the beginning of the flow decreases for modules M3 and M4, with respect to modules M1 and M2 (Fig. 2b). They coincide with the fast drop of the heat coefficient transfer (Fig. 1d) expected in an ideal case. It is possible to see a slight increase in temperature in module M3 before decreasing again. This is not so evident for module M4, which can be expected in data evaluation of modules exposed in the environment with natural winds, inherently turbulent, despite following similar patterns as expected for ideal cases. At the end of the flow, the developed turbulence maintains the temperature of the PV modules high due to the viscous sublayer in the turbulent zone. $\Delta T$ tends to be low with the heat transfer decrease in this zone (Fig. 2a). At the beginning of the flux, $\Delta T$ is higher due to the great heat transfer variation (six modules in the other extreme).

$\Delta T$ behaviour is almost identical both when the wind blows parallel or diagonally (front or rear side) in the PV panel (Suppl. Fig. S2a–c, Fig. 2a). This must be the case for all PV generators with a horizontal dimension much greater than a vertical one (with a few metres). In these cases, the vertical component influence of the wind must be depreciable. A higher temperature measurement resolution, capable of seeing slight temperature differences due to the inherently turbulent natural wind, is out of the scope of this work. When the wind blows perpendicularly to the front and the rear sides of the PV generator, the temperatures are well distributed along the PV generator, with similar $\Delta T$ in both extremes (Suppl. Fig. S2d, e).

The relevant information to be retained here is, despite the inherently turbulent nature of the wind, experimental observations show that the temperature distributions in a PV generator behave similarly as expected in an ideal case, which can be well explained by fluid mechanics properties with great accuracy.

**Impact of the wind speed and direction in the $MML_{DAY}$ values.** The rear wind incidence leads to lower $MML_{DAY}$ than the frontal wind incidence, as it tends to increase the heat transfer from the modules to the air. Frontal wind incidence, majorly from the South-West quadrant, leads to $MML_{DAY} = 0.28\%$, while rear wind incidence, majorly from the North, leads to $MML_{DAY} = 0.21\%$ (Fig. 3a, b). However, a decrease in wind speed also decreases the $MML_{DAY}$. Considering days with similar frontal and rear incidences, with less wind speed, a similar frontal wind incidence leads to $MML_{DAY} = 0.25\%$ (Suppl. Fig. S3a). For the rear side, $MML_{DAY} = 0.17\%$ (Suppl. Fig. S3b).

Furthermore, in these same lines, low wind speeds lead to low $MML_{DAY}$. For a day with very low or even null wind speed, $MML_{DAY} = 0.13\%$ (Fig. 3c), which means, in other words, a decrease of the energy losses with the wind speed decrease.

It is worth commenting that the case where $MML_{DAY} = 0.17\%$ must also be due to the absence of wind, reflected in the high frequency of null wind speed (Suppl. Fig. S3b). However, the high variability is also expected due to the great heat transfer coefficient variation in the rear side incidence of the wind[8], which must play an important role in the MML variation. This deserves further investigation.

**The thermal drop between the cell and the rear side under different wind incidences is the source of MML.** The thermal behaviour of the PV modules depends on their physical path, where the internal heat flows from the cell to the front or back surface. Because of that, a thermal drop correction[31] is widely

employed by many engineering applications. This implies that a constant thermal gradient between the cell and the rear side must exist. As the temperature widely varies inside the PV generator, depending on the local air flux behaviour, the thermal drop must also be affected due to the air flux variations. When the PV module is in the turbulent zone, the viscous sublayer imposes an additional thermal resistance between the module surface and the airflow, lowering the thermal gradient between the cell and the airflow and reducing the heat transfer in this zone. Thereby, the temperature increases, creating a thermal gradient between the PV cell and the rear side. Consequently, the heat flows towards this direction, exchanging temperature with the more stable air.

Considering the ideal case where the wind speed may increase the turbulence, as the transition point of the air flux regime moves towards its beginning, the heat transfer may be even lower and may affect in the same way other modules. This must be the reason behind the increase of the MML with the wind speed increase.

In Fig. 4a, the PV module (its location can be seen in Suppl. Fig. S4c) is the underdeveloped turbulent regime. $T_C^{VOC}$ remains higher than $T_C^{REAR}$, sometimes dropping for values below the latter with sudden wind speed increases. These temperatures remain about 20 °C above ambient temperature $T_{AIR}$. In Fig. 4b, with rear wind incidence, the thermal drop is even higher most of the time, especially when high wind speeds prevail, possibly with wind gusts. Both module temperatures are close to $T_{AIR}$, being in certain moments less than 10 °C above it. In Fig. 4c, the thermal drop between $T_C^{VOC}$ and $T_C^{REAR}$ is almost inexistent. In the absence of wind, PV temperatures remain about 30 °C above $T_{AIR}$.

The literature widely reports that cell temperature is usually 30 °C above ambient temperature $T_{AIR}$. This tends to occur in days prone to a steady state atmosphere, i.e., with very low or even null wind speed. $T_C^{VOC}$ and $T_C^{REAR}$ are about the same when the wind speed is very low or null (Fig. 4c). This must be the case in the whole PV generator in similar wind speed conditions, coinciding with the related lower MML. On the other hand, this difference decreases with the wind speed increase, reaching differences between 15 and 20 °C when the wind reaches the rear side of this specific module. Considering the entire PV generator, as the heat transfer is more variable with the rear wind incidence than the frontal incidence[8], the dispersion of the heat flux in the different parts of the whole PV generator must largely vary the thermal drop in its different zones. Taking into account the similarity of the days here analysed with the corresponding days where the MML analysis proceeds, these variations suggest a direct relationship between the internal heat flux discrepancy along the PV generator and the MML, which deserves further investigation.

A previous study suggested that the internal heat flux must be depreciable in comparison to the wind effects[32]. However, the results presented here suggest that both phenomena cannot be dissociated, as the heat flux in each part of the panel depends on the way that the wind interacts with it.

**Impact of the wind patterns in the $MML_{MONTH}$ variation.** The long-term losses of a PV installation may be understood through the $MML_{MONTH}$ estimations. These losses vary in the short term following the local wind patterns, which tend to be similar year-by-year, at least in the medium term. The natural ageing of the PV panel is responsible for the $MML_{MONTH}$ increase, which corresponds to 0.04% per year for the PV generator analysed here[1].

The local measurements presented in this work follow the local tendency, especially during most of the summer. For the specific PV system analysed here, South-oriented (Methods),

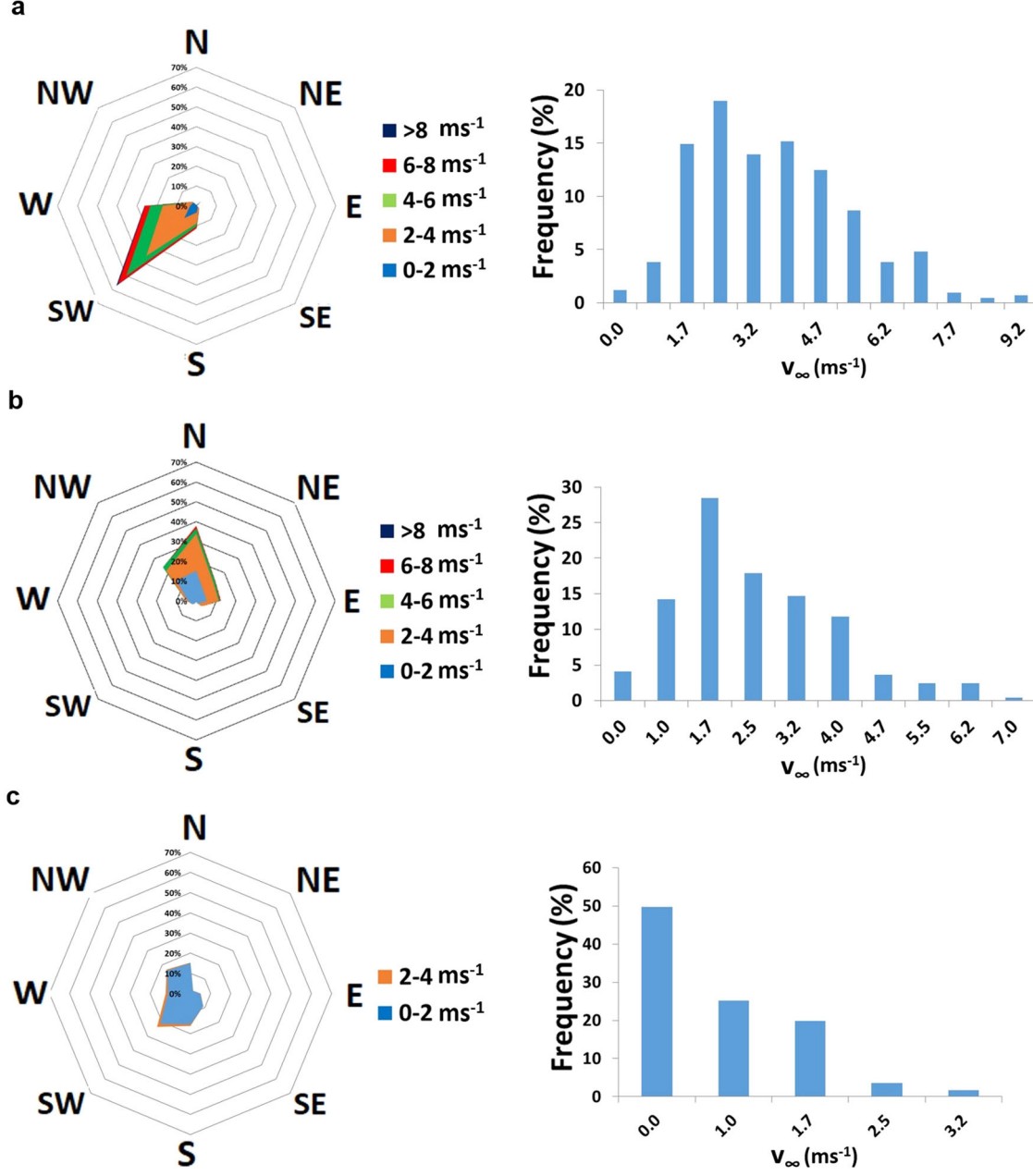

**Fig. 3 MML$_{DAY}$ variations with the wind patterns. a** Frontal wind incidence, high wind speeds, MML$_{DAY}$ = 0.28%. **b** Rear wind incidence, high wind speeds, MML$_{DAY}$ = 0.21%. **c** Single day with low or null wind speed, MML$_{DAY}$ = 0.13%. In each wind rose, N, NE, E, SE, S, SW, W and NW represent the North, North-East, East, South-East, South, South-West, West and North-West, respectively. MML stands for mismatch losses. MML$_{DAY}$ stands for daily mismatch losses, i.e. evaluated during one day.

MML$_{MONTH}$ increased in this period (Fig. 5a, b), coinciding with the thermal low over the Iberian Peninsula[33]. This can be seen in the three consecutive years presented in Fig. 5a. This increase in losses must occur for most of the PV systems with the same orientation located all over this region.

Wind from the South-West quadrant is relatively common for this location. This can be seen during three consecutive periods in 2018 (Fig. 6a–c). However, the wind from the other quadrants plays a more relevant role in the energy losses during January to March (Fig. 6a) and September to December (Fig. 6c), with more frequent low or even null wind speeds, coinciding with lower MML$_{MONTH}$ values. From June to August (Fig. 6b), higher wind speed from the South-West quadrant prevails, coinciding with the higher MML$_{MONTH}$ values.

It is worth noting that these same patterns are observed in the other years, which reinforces the connection between the local wind patterns with the MML.

## Conclusions

Here I reported the impact of the wind on the energy production of a PV generator currently injecting energy into the grid. The wind speed increase is responsible for increasing the energy losses. This apparent counter-intuitive argument follows the fluid mechanics theory, as the wind interaction with the PV generator induces air flux variations that modify the heat transfer from the modules to the air. Results suggest that the local wind patterns play an important role in estimating the losses: windy locations

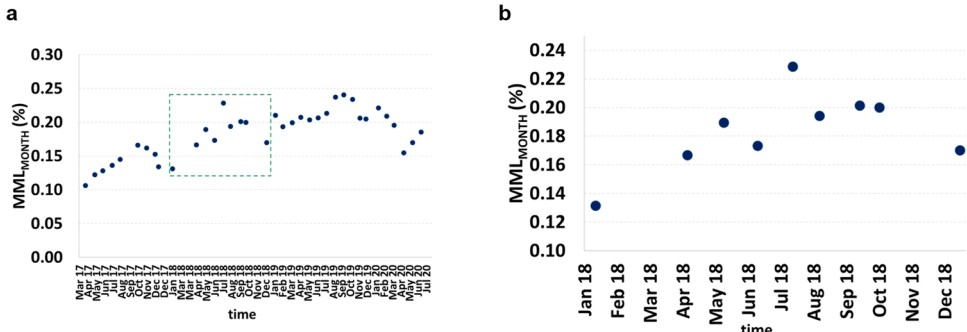

**Fig. 4 Thermal drop variations with the wind patterns.** Analyses relate to the calibrated module, depicting the respective wind roses, frequency distribution of wind speed and module (both rear and $T_{VOC}$) and air temperature variations with the wind speed ($v$). **a** Frontal wind incidence, mainly from West and South-West quadrants, with high wind speed values. **b** Rear wind incidence, mainly from North and North-West quadrants. **c** Day with prevailing low or null wind speeds. In each wind rose, N, NE, E, SE, S, SW, W and NW represent the North, North-East, East, South-East, South, South-West, West and North-West, respectively.

**Fig. 5 MML$_{MONTH}$ variations. a** MML$_{MONTH}$ increase during more than 3 years—from April 2017 to July 2020. Inside the green dashed rectangle are the values corresponding to 2018, as can be seen enlarged in (**b**). **b** MML$_{MONTH}$ during 2018, where the wind analyses took place. MML stands for mismatch losses. MML$_{MONTH}$ stands for monthly mismatch losses, i.e. evaluated during 1 month.

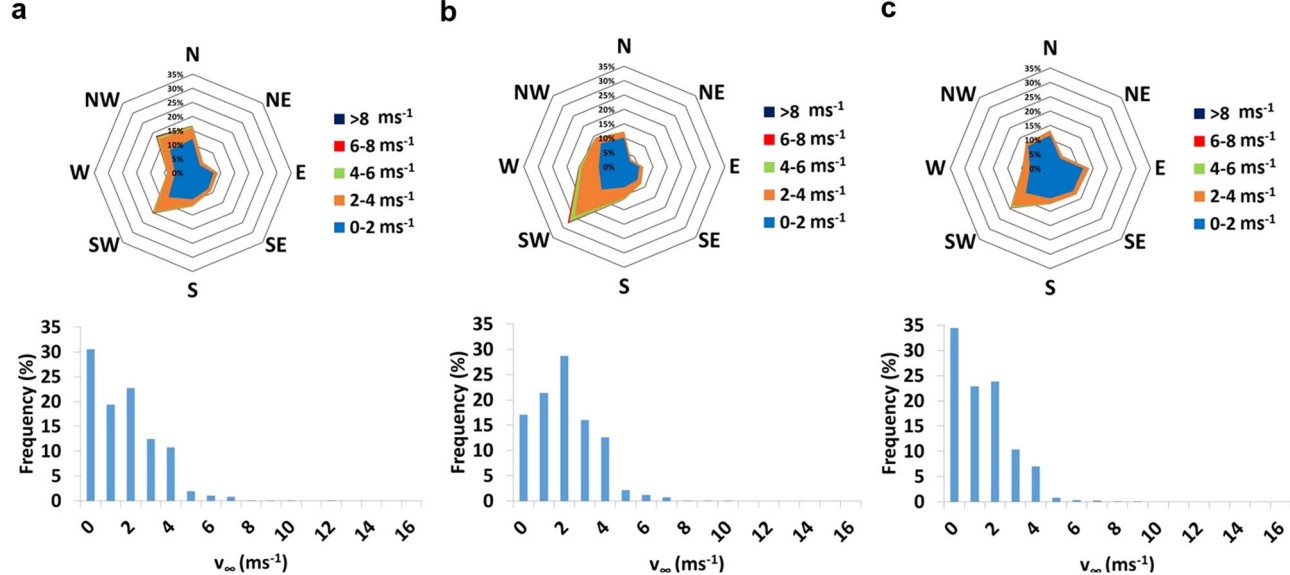

**Fig. 6 Wind rose and wind speed distribution for three distinct periods in 2018. a** January to March. **b** June to August. **c** September to December. In each wind rose, N, NE, E, SE, S, SW, W and NW represent the North, North-East, East, South-East, South, South-West, West and North-West, respectively.

lead to major energy losses, derived from the temperature differences, with major frontal incidences leading to higher losses than major rear incidences. In addition, low or null wind speeds lead to lower energy losses. The thermal behaviour that leads to these losses is intrinsically linked with the airflow properties, as they occur independently if the PV generator is injecting energy or not into the grid.

The thermal drop analysis between the cell temperature $T_C^{VOC}$ and the temperature measured in the back sheet of the module $T_C^{REAR}$ suggests that the ultimate reason for the MML increase must be related to the overall internal heat flux discrepancy in the whole PV generator between its different points, as a consequence of the wind interaction with it.

Considering that the consequences of the climatic change are occurring faster than expected years before[34,35], the increase of the heatwave frequency for the next decades in the Iberian Peninsula[36] must also affect the wind patterns behaviour. As the typical lifespan of a PV power plant may last some decades, this must represent an important uncertainty source to ensure the reliability of the PV plants. This suggests that the hitherto depreciated local wind patterns for energy estimations must be taken into account for a proper energy estimation during their lifespan. Along these lines, for locations with similar irradiance incidence, a less windy choice is preferable to reduce MML.

## Methods

**The PV generator: description and experimental assembly.** The analyses proceeded in one of the PV generators located at the Solar Energy Institute of the South Campus of the Technical University of Madrid—Spain (40.39°N, 3.63°W). It is composed of 21 PV modules Siliken SLK60P6L of 245 $W_P$ each, whose dimensions are $1.60 \times 0.99$ m, formed by 60 BSF multi-crystalline silicon cells connected in series. The PV generator consists of 3 parallel strings of 7 modules in series each, forming a rectangle of $10.89 \times 3.20$ m. An additional disconnected module closes the surface to avoid unexpected turbulence zones, which must interfere with the overall air flux. It is tilted 30°—practically the optimum angle for Madrid, Spain[37]. It is worth commenting that this is not necessarily the tilt angle on which the commercial solar PV systems are installed. The PV panel is South-oriented (azimuth = 0°) and has currently injected energy into the grid since

March 2013 (Suppl. Fig. S4a). The temperature and voltage are recorded through PT1000 sensors and T-shaped connectors (Suppl. Fig. S4d). The records comprise a database of more than 3 years of measurements, from February 25th 2017, to July 2nd 2020, with 5 min between each record, using a data logger with 20-bit digitalisation. All measurements proceed with the PV generator currently injecting electricity into the grid. The measurements are synchronised with a meteorological station, presented in the next section.

During some periods, the PV generator was not generating energy due to maintenance activities or other experiments, which was necessary to proceed with the measurements in open circuits. For the analyses of this work, these data were disregarded. Due to the shadows projected by a nearby building and a tree in the morning and in the afternoon, data were limited by a window free of shading varying from about 8h30min to 13h45min in winter and from about 7 h00 min to 14 h45 min in summer (both UTC). The procedures of the temperatures and operating voltage measurements are well described in a previous publication[1], which presents the method to estimate the MML from the operating voltage observations, briefly commented in this section. Additionally to the temperature measurements of the entirely PV generator, four modules placed at one of the extremes of the PV generator (M1–M4) have three PT1000 sensors glued in selected points of their rear side, as recommended by IEC 62853[38] (Suppl. Fig. S4c). The PT1000 sensors used are of class A, preferred in this kind of measurement due to their high accuracy in registering small temperature changes—±0.3 °C of uncertainty—, enough to assume the measured temperature, at least in this single point, as the real temperature.

The transient temperatures due to slight wind variations were measured in a single cell, with a thermographic camera FLIR E60 for 40 s between the 6 images. For a better visualisation of the cooling driven by the wind, the interval of temperature was set between 35 °C and 37 °C. Thus, the temperature decrease is understood as the absence of colour in the images. The images were taken on the 1st of February, 2022, at 11 h30 min (UTC).

One of these four modules (M3) (Suppl. Fig. S4c) was previously calibrated in a "solar box", an apparatus that allows the outdoor measurements of individual PV modules, controlling the incident irradiance and ambient temperature, to know their

values at standard test conditions (STC) and their thermal coefficients[39]. The irradiance that reaches the PV module and its cell temperature, $G$ and $T_C$ respectively, are deduced from the recorded $I_{SC}$ and $V_{OC}$ (now called $T_C^{VOC}$) through the following equations:

$$G = G^* \frac{I_{SC}}{I_{SC,REF}^*} \frac{1}{[1 + \alpha(T_C - T_C^*)]} \quad (2)$$

$$T_C^{VOC} = T_C^* + \frac{1}{\beta}\left(\frac{V_{OC}}{V_{OC,REF}^*} - 1 - a\ln\frac{G}{G^*}\right) \quad (3)$$

where $I_{SC}$ is the recorded short-circuit current, $I_{SC,REF}^*$ the calibrated $I_{SC}$ (for this module, $I_{SC,REF}^* = 8.44$ A), $\alpha$ is the thermal coefficient of short-circuit current (for this module, $\alpha = 0.057\%\ °C^{-1}$), $T_C^*$ is the cell temperature at STC (25 °C), $\beta$ is the thermal coefficient of open-circuit voltage (for this module, $\beta = -0.34\%\ °C^{-1}$), $V_{OC}$ is the recorded open-circuit voltage, $V_{OC,REF}^*$ is the calibrated $V_{OC}$ (for this module, $V_{OC,REF}^* = 36.99$ V), $a$ is the ratio of the thermal voltage to the open-circuit voltage ($a = 0.045$ for crystalline silicon modules) and $G^*$ is the irradiance in STC (1000 W m$^{-2}$). The $V_{OC}$ method for determining $T_C$ is suggested by IEC 60904-5[40].

The $T_C^{VOC}$ was compared with the measured temperature by the PT1000 sensor—$T_C^{REAR}$—under different wind incidences. These measurements proceeded on three different days, with similar wind patterns to the ones observed in the MML$_{DAY}$ analysis (Southwest incidence, North incidence and low or null wind speed). For this specific analysis, this module was in open-circuit, maintaining its same position inside the PV generator, which was also in open-circuit, with all the sunlight being converted into heat. This enlarges the thermal effects due to the wind, the scope of this analysis. The evolution of its $T_C^{VOC}$ and $T_C^{REAR}$ is then analysed alongside the air temperature $T_{AIR}$ at the same time, measured by a PT1000 sensor connected to the meteorological station.

**Meteorological data**. Close to the PV generator also installed in March 2013, is a Geonica™ meteorological station. It is currently recording, with a periodicity of 1 min, the wind speed and direction with an anemometer. A reference module from the same manufacturer and model of the modules that make up the PV generator is also connected to the meteorological station and is currently measuring both effective irradiance $G_{EF}$ and $T_C$[41] (Suppl. Fig. S4a, b). These measurements are synchronised with the measurements of voltage and temperature, which means that each 5 min of voltage and temperature measurements have their corresponding meteorological data measurement.

**Deriving MML from operating voltage dispersion**. As the current throughout the PV modules must be the same, the differences between them must be seen in their individual operating voltages. Assuming that the mean value of the operating voltage $\overline{V_{OP}}$ of all 21 modules corresponds to the operating voltage at the maximum power point, there are 21 $\Delta V_{OP}$ values, which correspond to 21 power deviations $\Delta P$ from the power at the maximum power point. The MML is derived then from the relative deviation CV$_{VOP}$, defined by the ratio between the standard deviation of $V_{OP}$ for its mean value, $\overline{V_{OP}}$, i.e., $\frac{\sigma_{V_{OP}}}{\overline{V_{OP}}}$. Both present a quadratic relationship, independently of the period of the year and can be expressed as follows:

$$\mathrm{MML} = a\,\mathrm{CV}_{VOP} + b\,\mathrm{CV}_{VOP}{}^2 \quad (4)$$

where $a$ and $b$ are adjustment coefficients experimentally

obtained for these specific modules ($a = 0.002$ and $b = 0.17$). This equation allows the MML calculation directly from the measured data with great accuracy[1].

The derived MML values are the sum of two components: intrinsic, which is related to the internal differences between the modules and can be considered constant in the short term, and extrinsic, which varies in the short term, even in a few minutes, and is related to the wind incidence that induces temperature differences in the PV generator due to air flux variations. In an ideal case, with a total absence of wind, the MML should be reduced to the intrinsic losses. As commented before, all analysed data is from the interval free of shading and during sunny days. It is then contrasted with the wind incidences and corresponding MML variations.

More details of the method and its development can be found in other previous publication[1].

**Reporting summary**. Further information on research design is available in the Nature Portfolio Reporting Summary linked to this article.

## Data availability
The datasets generated during and/or analysed during the current study are available from the corresponding author upon reasonable request.

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

## Acknowledgements

I wish to thank the PV Systems Group from the Solar Energy Institute—Technical University of Madrid for the support in the voltage and temperature measurements and the development of the method to estimate the mismatch losses. I also wish to thank CNPq (Conselho Nacional de Desenvolvimento Científico e Tecnológico), which funded this work during the measurements stage. I thank the Climate Physics Group from the University of Alcala, in the person of Professor William Cabos, who believed in the project and opened the doors for its development, giving insightful ideas. Finally, I am very grateful to Angelita Meirelles de Lima da Silva, for believing in the project and its significance and to Mariana Heinz, whose insights, comments and support were fundamental during the development of this study.

## Author contributions

C.R.: investigation, experiments, data collection and analysis, theoretical analysis, bibliographic analysis, conceptualisation, methodology, visualisation and writing (original draft, review and editing).

## Competing interests

The author declares no competing interests.
