## [Peer Review File · Communications Engineering]

Reviewers' comments:

Reviewer #1 (Remarks to the Author):

The paper needs some major revisions.

It needs to be better organized to help the reader understand. The introduction reads more like a methods section, but part of the method seems at the beginning, part is at the end.

All figures are small and difficult to read.

I also have some concerns: Extended Fig. 4 a shows the PV system. At the left of the image is some vegetation/pole(?) visible. You can also see the shadow on the panels. How can you know that the difference you detect are due to wind changes and not due to shading?

Module temperatures typically have a +/- 2C uncertainty associated with them, the uncertainty of these measurements need to be discussed too.

Reviewer #2 (Remarks to the Author):

“operating voltage and temperature measurements in each module of a PV generator”

Please describe the size, mounting and layout of the PV plant. Especially, the string length and layout of modules within each string. Please provide image(s).

“temperature dispersion” – please define

“*MML* of 0.09%”

Please define/explain what this % figure means practically

“The thermal behavior in larger systems follows the fluid mechanics theory for flat plates”

Temperature distribution of PV modules also strongly depends on their rack/mounting structure, especially for tracker-mounted bifacial modules (the industry standard). Please consider and discuss.

“The method of *MML* analysis used here”

Please clarify if you are talking about mismatch between cells within a module, or between modules within a string.

“operating voltage dispersion ΔV_{OP} ” – please define

“shunt resistance dispersions” – please define. In general, “dispersion” in English is commonly used to mean a spreading out, so the way you tend to use it is not clear. Please consider an alternative term or clearly define what phenomenon you are describing.

“While the wind flows throughout the panel, both the velocity and thermal boundary layers grow”

It seems that your model assumes that the module is parallel to the free-stream airflow. If so, this is a highly unrealistic assumption, as PV modules are usually tilted (and one would by default assume roughly horizontal wind). Please clarify this point and clearly state the major assumptions upon which

your model relies.

It seems your model assumes that wind is laminar until a turbulent boundary layer develops some distance from the leading edge. This assumption is not realistic, according to the literature natural wind is inherently turbulent. Please clarify and discuss this assumption.

“This viscous under-layer acts as an extra path to the heat flow, which explains why the temperature of the modules remains high in this zone”

It would seem that a simpler explanation is that the entire boundary layer thickens in the x direction, reducing the flow velocity near the surface. The viscous sublayer exists at all locations – even immediately after the leading edge - and is of uniform thickness, so it does not explain the variation in heat transfer in the x-direction

Your results seem to assume that the wind direction is parallel to the PV string. This seems like a highly special and uncommon situation. Please consider and discuss the realism of this assumption.

It is stated that this is an experimental study, but no details of the test set-up/location are provided. Please complete this.

Reviewer #3 (Remarks to the Author):

The paper concerns mismatch losses in a PV generator considering different wind patterns.

The manuscript is well-written and well-organized. However, there are some important issues I believe you may need to follow before the paper is ready to publish.

1- the quality of all figures is very low, it is almost impossible to read the text inside the figures and plots. Please redraw all with bigger font text.

2- The manuscript refers to the PV generator system which is a rooftop system depicted in Ext. Figure 4. It is not indicated the orientation of the system, which is important to know. In line 150 it is mentioned that the wind coming from the North is more effective in temperature drop which is expected for a system with the south orientation and I do not consider it a big finding in a research project. Please indicate the system orientation.

3- The system tilt angle is 30, I would like to know why the author chose this angle? Although the optimum angle for the location of the research might be equal to 30, is that an angle industrial parties would install the panels with, if not how can you address the losses with different tilt angles?

4- There is no definition for mismatch losses. Please make it clear how did you recognize the losses are affected by wind regime/speed change in the practice tests?

5- In lines 49-54 the variation of voltage with respect to irradiation change is practically described and refers to recorded data and a scattered plot in Extended data fig 1. Please provide numerical/mathematical proof for this.

6- the cross-references are not well-organized, please consider there are subplots for each figure, please refer correctly.

Dear reviewer,

Thank you for your useful remarks regarding the paper entitled “Energy losses in photovoltaic generators due to wind patterns”. Your comments certainly improved the quality of the work, improving its readability. You can see below each comment and/or answer addressing each specific remark.

Reviewer #1 (Remarks to the Author):

The paper needs some major revisions.

It needs to be better organized to help the reader understand. The introduction reads more like a methods section, but part of the method seems at the beginning, part is at the end.

The text was better organised to fix this and allow the correct understanding of the text.

All figures are small and difficult to read.

Figures were enlarged (images or axes when applied) to allow the better visualisation.

I also have some concerns: Extended Fig. 4 a shows the PV system. At the left of the image is some vegetation/pole(?) visible. You can also see the shadow on the panels.

How can you know that the difference you detect are due to wind changes and not due to shading?

All data were evaluated in an interval free of shading, as commented in the section Methods (lines 242 to 244). In fact, the photo can induce some misinterpretation due to the projected shadows. To correct it and avoid some other possible misinterpretations, I changed to another photo free of shadings, took from a similar angle.

Module temperatures typically have a +/- 2C uncertainty associated with them, the uncertainty of these measurements need to be discussed too.

The uncertainties of temperature measurements address specifically to the related PT1000 sensor uncertainty and the datalogger used during the experiment (commented in lines 249 and 448, the last as an added comment). For that, the following sentence was added:

In lines 249 to 251, to specify the class of the temperature sensor: “The PT1000 sensors used are of class A, preferred in this kind of measurement due to their high accuracy in registering small temperature changes - $\pm 0.3^{\circ}\text{C}$ of uncertainty -, enough to assume the measured temperature, at least in this single point, as the real temperature.”.

The $\pm 2^{\circ}\text{C}$ of module temperature associated uncertainty you commented is essentially related to the procedure to obtain the real temperature than the temperature itself, if I properly understand your comment. If it is so, the “real module temperature” is commonly assumed as the temperature calculated from its Voc, as suggested by IEC 60904-5. This method was used in the single module analysed in the chapter “The thermal drop between the cell and the rear side under different wind incidences as the source of MML”. For the other modules, the experimental setup was limited to measurements in single points, as can be seen in Extended Data Fig. 4 c. However, it suffices for the purposes presented in the article. The maximum temperature differences that emerge inside modules and the generators, which can be understood as the source of the uncertainty to know the real temperature, are commented in lines 47 to 51.

Dear reviewer,

Thank you for your useful remarks regarding the paper entitled “Energy losses in photovoltaic generators due to wind patterns”. Your comments certainly improved the quality of the work, improving its readability. You can see below each comment and/or answer addressing each specific remark.

Reviewer #2 (Remarks to the Author):

“operating voltage and temperature measurements in each module of a PV generator”
Please describe the size, mounting and layout of the PV plant. Especially, the string length and layout of modules within each string. Please provide image(s).

These details are already specified in the section “Methods”, between the lines 229 and 235. The “Extended Data Fig. 4|Experimental setup” already provided the images of the PV generator. The Fig c was modified to specify each of the three strings with their seven modules connected in series.

“temperature dispersion” – please define

As you correctly pointed in other remark, it seems that “dispersion” is not properly employed at all in some of the contexts intended here. Therefore, I changed “dispersion” to “difference”, when applied. In this case, the temperature difference is properly introduced with the following added sentence, between lines 37 and 39: “However, this masks the effects of air flux variations and consequent temperature differences ΔT – i.e. the difference between the maximum and minimum values of measured temperatures - inside them, which affects the electrical parameters inside each PV module.”.

“*MML* of 0.09%”

Please define/explain what this % figure means practically

Complementing the idea in lines 7 and 8, and connecting to the explanation of these losses in the absence of thermal variations, the following sentence was added, in lines 8 to 10: “This means that, according to its series/parallel configuration, electrical characteristics and their operating conditions dispersion, the minimum expected losses for this generator are of 0.09%.”.

“The thermal behavior in larger systems follows the fluid mechanics theory for flat plates”

Temperature distribution of PV modules also strongly depends on their rack/mounting structure, especially for tracker-mounted bifacial modules (the industry standard).

Please consider and discuss.

The temperature distribution is a direct consequence of the airflow in flat plates, according the theory and the results presented in the article. However, as you commented, it was not clear the effect on these temperatures due to the rack/mounting structure. I complemented the idea with your comment, explaining

that the mounting configuration influences on how the wind interacts with the PV panel and affects the temperature distribution. I also added a comment on a previous study reporting a difference of temperature distribution between horizontal and vertical dimensions in a PV generator with similar dimensions, briefly explaining their results under the light of the results presented here. I expect this comment reinforces the idea well suggested by you with an example already published with experimental observations. You can see the added comment between the lines 16 and 23, which I reproduce here: “This is also affected by the rack/mounting structure, especially for tracker-mounted bifacial modules, as they affect how the incident wind flows throughout the panel and, consequently, how the heat is transferred from the surface to the air. These differences were reported in a previous study - up to 14 °C -, with an analysis of a PV generator with an azimuthal tracker, tilted 45°, with similar vertical and horizontal dimensions, whose temperature distribution presents difference between the two dimensions. The tilt causes an acceleration of the wind incident, increasing the turbulence and reducing the maximum temperature difference ΔT between two measurement points, which explains the lower ΔT in the vertical dimension than in the horizontal, as reported by this study”.

“The method of *MML* analysis used here”

Please clarify if you are talking about mismatch between cells within a module, or between modules within a string.

To clarify this point, I specified in the first sentence of the paper – lines 3 and 4 – what the analysis stands for specifically, as a matter of introduction before to discuss the intrinsic and extrinsic components. Here I reproduce the sentence: “This paper analyses the energy losses in photovoltaic (PV) generators due to the wind patterns, assessed through the experimental mismatch losses (*MML*) analysis between PV modules.”. Besides, I complemented the idea in the sentence you commented, between lines 39 and 41, as follows: “The method of *MML* – see section Methods - analysis between PV modules used here is accurate enough to see these slight variations due to the wind speed and direction.”.

“operating voltage dispersion ΔV_{OP} ” – please define

Here is the other case where “dispersion” was not correctly employed at all – same as in the case of temperature as you correctly pointed out. Thus, here I changed to “difference” too, and in all the cases when applied. In lines 52 and 53, the definition of operating voltage difference can be seen: “The operating voltage difference ΔV_{OP} – i.e. the difference between the maximum and minimum values of measured operating voltages”.

“shunt resistance dispersions” – please define. In general, “dispersion” in English is commonly used to mean a spreading out, so the way you tend to use it is not clear. Please consider an alternative term or clearly define what phenomenon you are describing.

In this case, “dispersion” is properly employed. The misunderstanding was probably due to the cases where the same word was used to define maximum differences between operating voltages and temperatures. In lines 63 and 64, it is now specified that the comment refers to the “Extended Data Fig. 1 e”.

“While the wind flows throughout the panel, both the velocity and thermal boundary layers grow”

It seems that your model assumes that the module is parallel to the free-stream airflow. If so, this is a highly unrealistic assumption, as PV modules are usually tilted (and one would by default assume roughly horizontal wind). Please clarify this point and clearly state the major assumptions upon which your model relies.

The section introduces a theoretical explanation behind the observed temperature distributions experimentally observed. To emphasise that this is an idealisation to understand the following results, the phrases between lines 66 and 68 were slightly changed, adding the information that conducts the reader to the theory. The sentence can be seen here: “To understand the thermal behaviour in a PV panel, it can be first assumed that the wind flows parallel to it. According to the theory, for flat plates, the thermal gradient between the surface and the air strongly depends on the conditions of the thermal boundary layer.”.

The following lines of the chapter were adjusted to emphasise the observed results with the theoretical explanation when applied. The section ends with an additional sentence, reinforcing that the experimental results can be interpreted with great accuracy as a consequence of fluid mechanics properties (lines 118 to 120).

It seems your model assumes that wind is laminar until a turbulent boundary layer develops some distance from the leading edge. This assumption is not realistic, according to the literature natural wind is inherently turbulent. Please clarify and discuss this assumption.

I added in lines 99, 105, 108 and 120 the term “ideal case”, to avoid possible misunderstanding to the reader that I am comparing the observed results with the theory, reinforcing, as explained in the previous remark, that this paragh is giving a brief theoretical explanation to understand the observed results.

“This viscous under-layer acts as an extra path to the heat flow, which explains why the temperature of the modules remains high in this zone”

It would seem that a simpler explanation is that the entire boundary layer thickens in the x direction, reducing the flow velocity near the surface. The viscous sublayer exists at all locations – even immediately after the leading edge - and is of uniform thickness, so it does not explain the variation in heat transfer in the x-direction

Between lines 76 and 81, I made some changes in the text, taking into account your suggestions and your useful comment concerning the phenomenon.

Your results seems to assume that the wind direction is parallel to the PV string. This seems like a highly special and uncommon situation. Please consider and discuss the realism of this assumption.

As commented before in one of your remarks, the parallel flow was used as a theoretical explanation to introduce the experimental observations.

It is stated that this an experimental study, but no details of the test set-up/location are provided. Please complete this.

Details were already provided in section methods on the first version of the article. This section can now be seen from lines 226 to 308, just before the references.

Dear reviewer,

Thank you for your useful remarks regarding the paper entitled “Energy losses in photovoltaic generators due to wind patterns”. Your comments certainly improved the quality of the work, improving its readability. You can see below each comment and/or answer addressing each specific remark.

Reviewer #3 (Remarks to the Author):

The paper concerns mismatch losses in a PV generator considering different wind patterns.

The manuscript is well-written and well-organized. However, there are some important issues I believe you may need to follow before the paper is ready to publish.

1- the quality of all figures is very low, it is almost impossible to read the text inside the figures and plots. Please redraw all with bigger font text.

Figures were enlarged (images or axes when applied) to allow the better visualisation.

2- The manuscript refers to the PV generator system which is a rooftop system depicted in Ext. Figure 4. It is not indicated the orientation of the system, which is important to know. In line 150 it is mentioned that the wind coming from the North is more effective in temperature drop which is expected for a system with the south orientation and I do not consider it a big finding in a research project. Please indicate the system orientation.

Regarding the orientation of the PV panel, it was already mentioned in line 234 (‘South-oriented’), alongside its tilt. I also add the geographical coordinates where the PV generator is located, in line 229, despite having in the previous version of the article the description of its location. Thus, reinforcing through the coordinates that it is located in the northern hemisphere, it is easily deductible that it is oriented towards the South. Besides, I added the azimuth (0°) to avoid any misinterpretations.

Regarding the remark on line 150, I changed a bit the sentence. It did not intend to represent more than a simple comment about the figures. However, as you correctly pointed out, the comment could lead to this mistake. The changes can be seen now between lines 171 and 174.

3- The system tilt angle is 30, I would like to know why the author chose this angle? Although the optimum angle for the location of the research might be equal to 30, is that an angle industrial parties would install the panels with, if not how can you address the losses with different tilt angles?

The PV panel was assembled with tilt angle of 30° to maximise the annual energy production for the location. The annual mean of daily irradiation in Madrid for a fixed surface occurs around this tilt, as can be seen in the reference added in line

234, commenting the tilt: “It is tilted 30° - practically the optimum angle for Madrid, Spain”.

The losses in different angles are not specifically addressed in this article, as they could not be experimentally analysed. However, based on the airflow behaviour commented on in the article, the tilt does not seem to present great influence in the MML for most of the actual PV installations, due to the low vertical dimensions respect to the horizontal ones – which tend to accelerate the wind. The MML due to the wind patterns is linked with the temperature differences inside the PV generator, which can be evaluated in these two dimensions. More details can be found between lines 18 and 25. On the other hand, considering just the horizontal dimension – and the horizontal component x of the wind – the tilt does not influence the flux. More detailed observations, e.g. long-term infrared observations, which are out of the scope of this article, must see slight differences in temperature distribution. For the present purposes, the procedure of temperature measurements suffices to explain the MML variations due to the wind patterns with good agreement with the expected behaviour predicted by the theory in an ideal case, even with the inherent turbulence of the natural wind.

4- There is no definition for mismatch losses. Please make it clear how did you recognize the losses are affected by wind regime/speed change in the practice tests?

I added in the section Methods (lines 226 to 309) a brief explanation of the MML derivation from the operating voltages. As the method is already explained in a previous publication, I opted to write the main idea to understand the MML in this paper, referencing the original publication for more details about the method. It is also explained how the wind can affect the MML variations. This is also explained in the introduction (lines 4 to 14), as it introduces the concepts of intrinsic and extrinsic MML components.

5- In lines 49-54 the variation of voltage with respect to irradiation change is practically described and refers to recorded data and a scattered plot in Extended data fig 1. Please provide numerical/mathematical proof for this.

It is described here the observations as a matter of introduction to the MML from the operating voltage. Based on your comments, I properly addressed the comments as a first approach to understanding the phenomenon, giving a first explanation for what will be seen in detail in the sections after. With the additional comments already suggested by you and other reviewers, it must be clearer for the reader. The lack of references was fixed in the comment concerning the high ΔV_{OP} in low irradiances, being one of them the experimental measurements of shunt resistance of the modules of this article. You can see the changes between lines 47 and 62.

6- the cross-references are not well-organized, please consider there are subplots for each figure, please refer correctly.

Corrections made, addressing including the section in some cases, when applied, where the text refers to a previous figure.

REVIEWERS' COMMENTS:

Reviewer #1 (Remarks to the Author):

Thank you for the clarifications.

Reviewer #2 (Remarks to the Author):

Thank you for addressing my previous comments

Reviewer #3 (Remarks to the Author):

The revised manuscript is improved a lot. However, it should be mentioned in the article that all discussions are based on the optimum tilt angle for that specific location which is not necessarily the tilt angle on which the commercial solar PV systems are installed.

Reviewer #3 (Remarks to the Author):

The revised manuscript is improved a lot. However, it should be mentioned in the article that all discussions are based on the optimum tilt angle for that specific location which is not necessarily the tilt angle on which the commercial solar PV systems are installed.

Thank you for your comment. Addressing your remark, I added the following sentence in the section Methods, first paragraph, accordingly your suggestion: “It is worth commenting that this is not necessarily the tilt angle on which the commercial solar PV systems are installed.”

Regarding this specific remark, I added reviewer's suggestion in the first paragraph at the section Methods, as follows: (...)It is tilted 30° - practically the optimum angle for Madrid, Spain. It is worth commenting that this is not necessarily the tilt angle on which the commercial solar PV systems are installed. The PV panel is South-oriented (...). The sentence simply acts as an additional information about the tilt. As commented in the review process, the present purposes do not address any other tilt, even if almost certainly the tilt does not have a significant influence on these losses for most commercial PV plants - mainly with the horizontal dimensions much greater than the vertical one.